# Hair Follicle Development and Cashmere Traits in Albas Goat Kids

**DOI:** 10.3390/ani13040617

**Published:** 2023-02-09

**Authors:** Xiaogao Diao, Lingyun Yao, Xinhui Wang, Sen Li, Jiaxin Qin, Lu Yang, Liwen He, Wei Zhang

**Affiliations:** 1State Key Laboratory of Animal Nutrition, College of Animal Science and Technology, China Agricultural University, Beijing 100193, China; 2Department of Animal Nutrition and Feed Science, College of Animal Science and Technology, China Agricultural University, No. 2, Yuan Ming Yuan West Road, Beijing 100193, China

**Keywords:** cashmere goat, cashmere, secondary hair follicle, *FGF*, *BMP*

## Abstract

**Simple Summary:**

The development of secondary hair follicles in goat kids determines cashmere quality in adults; however, this requires theoretical support. After a follow-up trial, we found that the development of secondary hair follicles in kids is complete at 5–6 months of age. *FGF*2, *21* and *BMP*7 may promote the growth of secondary hair follicles; and the optimal period for secondary hair follicle regulation is from birth to 5–6 months of age. In addition, hair follicle traits at 6 months of age can be used as an index for the breeding selection of cashmere goats.

**Abstract:**

The objectives of this trial were to study the growth and development of hair follicles and cashmere traits in cashmere goats and to provide a theoretical basis for the regulation of secondary hair follicle development and the scientific breeding selection of cashmere goats. Twelve single-fetal female kids were selected as research objects. A long-term tracking plan was created to regularly determine their growth performance, cashmere performance, and hair follicle traits. The results showed no significant difference in live weight after the first and second combing. The cashmere yield and unit yield of the first combing were significantly higher than those of the second combing (*p* < 0.05). Sections of hair follicles showed that the primary hair follicles are almost fully developed by 1 month, and the secondary hair follicles are fully developed by 5–6 months after birth. The primary hair follicle density (PFD) and secondary hair follicle density (SFD) were highest at birth and decreased within 1 month; and SFD was stable at 5–6 months of age. The change of MSFD took a maximum time of 2 to 3 months. The S:P increase reached its peak at 6 months. *BMP*4 expression increased with time. *FGF2*, *FGF21* and *BMP7* were higher at 3 months old than at the other two-time points. In conclusion, this study determined the total development time of primary and secondary hair follicles from morphology and speculated that *FGF2*, *FGF21*, and *BMP7* may play a regulatory role in developing secondary hair follicles. Therefore, the period from birth to 6 months of age was the best time to regulate secondary hair follicle development in cashmere goats kids. The traits of the hair follicle and cashmere at 6 months of age could be breeding selection indicators for cashmere goats.

## 1. Introduction

China is the world’s largest producer, processor, and marketer of cashmere; with total cashmere production accounting for over 70% of the world’s total production. China also has excellent cashmere goat varieties, of which the Inner Mongolia Albas cashmere goat is renowned for its high cashmere yield and good quality [1,2]. The quality of cashmere is determined by its yield, fineness, and length [3]. The diameter of Inner Mongolia Albas cashmere is mainly between 14 and 16 μm, and the super fine variety diameter was 14.5 μm or less. Cashmere quality is affected by genetic, environmental, and feeding management factors. Some studies show that cashmere yield is correlated with live weight, length, and fineness [4,5]; moreover, cashmere yield can be increased under supplementary feeding [6,7].

The hair follicle is a skin-derived organ and is divided into primary and secondary follicles, with the primary hair follicle growing coarse hair and the secondary hair follicle growing cashmere [8,9]. The growth of cashmere is cyclical, emerging from the body in August each year and shedding naturally in April of the following year. Cashmere growth and shedding are determined by the cycle of the secondary hair follicles, which consists of the anagen, catagen, and telogen phases [10,11]. The growth of secondary hair follicles is a complex process regulated by a variety of signaling pathways, of which the *FGF* and the *BMP* signaling pathways are well studied [12]. The fibroblast growth factor (*FGF*) family is expressed in human skin. *FGF1*, *2*, *5*, *7*, *10*, *13*, and *22* are known to be expressed in dermal and hair follicular cells and regulate hair growth and skin regeneration [13]. *FGF2* can promote the proliferation of the central neuron cells of mesenchymal cells and bone marrow mesenchymal stem cells [14]. *FGF21* is involved in hair follicle development and promotes secondary follicle development in the retrograde period [15]. Bone morphogenetic proteins (*BMPs*) belong to the transforming growth factor-β (*TGF-β)* superfamily, and *BMPs* are involved in a wide range of biological processes, such as cell proliferation, differentiation, bone formation, tumors, and hair follicle growth [16]. It is generally believed that *BMPs* have inhibitory and antagonistic effects on hair follicle growth, and *BMP2*, *BMP4*, *BMP6*, and *BMP7* have been identified as candidate genes for hair follicle development [17,18,19]. Zhang et al. proposed a model to explain a mechanism controlling expression of the *BMP4* gene in different tissue types, as well as different development stages as related to hair development [20]. *BMP7* is highly expressed in the growth phase of hair follicles and is low or undetected in the degenerate or resting phase [21]. Lv et al. noted that *BMP7* may promote the proliferation of hair follicle cells in the hair follicle growth phase [22]. Noramly et al. found that *BMP7* was associated with the size and spatial distribution of the feather germs, implying a potential role in hair follicle growth and development [23]. Furthermore, previous RNA-seq data from our study of adult goats and kids found that *FGF2/21* and *BMP4/7* were highly expressed (Table 1). Therefore, we speculated that *FGF2/21* and *BMP4/7* played an important role in the skin hair follicles of goats.

In addition, the breeding selection of cashmere goats includes two stages: (1) the choice of goat kids based on their weight and (2) the choice of adult goats based on their weight and cashmere traits. Nonetheless, in neither case are secondary hair follicle traits considered. Moreover, we speculate that young age is the critical period for regulating the development of secondary hair follicles, as it determines the “Lifetime cashmere quality”. Therefore, this study observed the morphogenesis of hair follicles in goat kids and regularly determined the secondary hair follicle traits and expression of crucial genes in order to provide a theoretical basis for regulating the development of secondary hair follicles, and to create a reference for scientific breeding selection of Inner Mongolia cashmere goats.

## 2. Materials and Methods

### 2.1. Animals and Sampling

This study was conducted on a commercial farm (YiWei white cashmere goat farm) located in the Inner Mongolia Autonomous Region, China, 39°06′ N, 107°59′ E. Twelve female, singleton, Inner Mongolian Albas cashmere goat kids, which were healthy newborns with a mean live weight (3.00 ± 0.56), were selected as the experimental subjects. Long-term follow-up studies were conducted (at ages 1 day to 6 months, 12 months, and 24 months, respectively) with regular weight, skin, and cashmere samples. Four skin samples were collected regularly at the posterior edge of the left scapula and the upper 1/3 of the line between the middorsal line and the midabdominal line for the skin test. Histological sections were stained with Sacpic method and then they were accurately analyzed under a photomicroscope (Leica ICC 50 W, Leica, Wetzlar, Germany) to determine the traits of the hair follicles [24,25]. The hair follicle index included the primary hair follicle density (PFD), secondary hair follicle density (SFD), mature secondary hair follicle density (MSFD), S:P, and the ratio of MSF. The total cashmere weight of each dam was recorded after combing, and cashmere samples within an area of 10 × 10 cm were shorn from the left midside of each dam close to the skin for the measurement of fiber staple length and diameter. The procedures for determining hair follicle indexes and cashmere traits refer to Yang and Duan [26,27].

### 2.2. RT-qPCR

*FGF*2, *FGF21*, *BMP4*, and *BMP*7 mRNA expression levels in the skin were measured using Q-PCR. The total RNA from the skin tissue was extracted using TRIzol (Takara, Dalian, China). All primer sequences were determined using established GenBank sequences, which are listed in Table 2. RNA was reverse transcribed to cDNA using a cDNA Synthesis Kit (Servicebio, Beijing, China). Real-time qPCR was performed on an ABI7500 system (Applied Biosystem, CA, USA) using a qPCR Detection kit (SYBR Green) (Servicebio, Beijing, China). Data were processed using ABI7500 Software v. 2.0.6 (Applied Biosystem, CA, USA). The RT-PCR analysis was performed using the 2^−ΔΔCT^ method.

### 2.3. Immunofluorescence

Skin specimens were fixed in 4% paraformaldehyde and were then dehydrated and embedded in paraffin. Paraffin sections were successfully dewaxed in different gradients of the dewaxing solution, absolute ethanol, and distilled water. The dewaxed sections were placed in EDTA antigen repair buffer (pH 8.0) for antigen repair and blocked with a 10% donkey serum block. Primary antibodies were then added: anti-rabbit-*FGF-21* (1:200) and *BMP*7 antibody (1:200), and incubated overnight at 4°C. On the next day, the slides were washed three times with PBS (pH 7.4) in a rocker device, the objective tissue was covered with a secondary antibody (responding appropriately to the primary antibody in the species), and was incubated at room temperature for 50 min in dark conditions. Finally, DAPI dye solution was added and incubated for 10 min. The images were observed and collected under a fluorescence microscope (GE Bioscience, Newark, NJ, USA).

### 2.4. Statistical Analyses

We used Excel to collect and organize the data, and the SPSS28 (IBM, New York, NY, USA) general linear model was used to compare the hair follicle indicators and cashmere performance in each stage. The results were expressed as mean ± standard error. A difference was considered significant at *p* < 0.05.

## 3. Results

### 3.1. The Live Weight of Cashmere Goat at Different Phases

We tracked the weight of cashmere goats within two years of age (Table 3). The weight of goat kids increased steadily from birth to 6 months, reaching nearly 18 kg, and remained at approximately 28 kg at the first and second combings (*p* < 0.05). The same weight provides a good analysis environment for us in the statistics of cashmere and hair follicle traits.

### 3.2. Cashmere Performance at Different Phases

We measured the cashmere characteristics after weaning at 3 months of age (Table 4). The diameter at 6 months (12.80 μm, *p* < 0.05) was significantly lower than that at other time points. The diameter was approximately 13.50 μm and the length was approximately 9.10 cm at the first and second combing. The yield and the yield per unit live weight of the first combing (869.48 g, 30.58 g/kg, *p* < 0.05) were significantly higher than those from the second combing.

### 3.3. The Development of Hair Follicles at Different Phases

Figure 1 shows the cross and longitudinal sections of hair follicles from the 1st day to the 180th day. As seen in the cross-section, most primary hair follicles completed development on the 1st day, whereas some had not yet. However, the secondary hair follicles were immature and in small number. The hair follicle morphology shows purple cell clusters in the differentiation stage. At 1 month old, primary hair follicles were thoroughly developed and the number of mature secondary hair follicles increased rapidly; and the distance between hair follicles was relatively loose. At 5–6 months of age, the morphology and the number of secondary hair follicles tended to be stable. Longitudinal and cross-sections showed similar results. Nonetheless, an interesting observation was that secondary hair follicles grew to run downward from the skin surface over time and were closest to the primary hair follicle in the same follicle group at 5–6 months.

### 3.4. Hair Follicle Traits at Different Phases

Table 5 shows the long-term record of hair follicle traits. PFD decreased gradually from day 1 to 6 months (9.97–3.86 n/mm²) and stabilized at between 5 and 6 months and 1.5 years old (the middle period of the second anagen). Furthermore, there was no difference between 4 months and 5 months, 6 months, and 1.5 years old (*p* > 0.05). The changing trend of SFD was similar to that of the primary hair follicles (60.75–46.83 n/mm²). The SFD at 6 months to 1.5 years was similar (*p* > 0.05). MSFD increased rapidly from 1 month of age, gradually approaching the number of total secondary hair follicles; and the change in MSFD occurred at a maximum of 2 months to 3 months of age, suggesting that the growth of secondary hair follicles is very active at this time. The ratio of MSFD was approximately 0.5–0.7 prior to 2 months of age and gradually levelled off with increasing age, approaching 1.0. S:P increased at the age of 1 month (6.10–12.64) and at 6 months; and the maximum value was not significantly different from that at the 4th and 5th months, and 1.5 years old (*p* > 0.05).

### 3.5. The mRNA and Protein Expression of BMP and FGF

The expression of *FGF* and *BMP* genes on the 1st, 90th, and 180th day-olds was tested (Figure 2). We found the expression of *FGF2*, *21*, and *BMP*7 genes on the 90th day was higher than that at the other two time points. The level of *BMP4* expression increased over time. Moreover, the protein of *FGF*21 and *BMP*7 was determined (Figure 3). *FGF21* protein was expressed mainly in the dermal papilla, inner root sheath, outer root sheath, and epidermis. *BMP7* was mainly expressed in the interstitial tissue between hair follicles, almost not at all in the hair follicle area; and the expression level of *BMP*7 in the epidermis was lower than that of *FGF21*.

## 4. Discussion

In this study, the growth of hair follicles and cashmere on Inner Mongolia Albas cashmere goats from birth to 2 years of age was tracked. We found that the primary and secondary hair follicles were developed entirely by 1 month and 5–6 months of age, respectively. The cashmere and unit yields of the first combing were higher than those of the second combing. The expression levels of the *FGF*2, *21*, and *BMP*7 genes-essential genes for hair follicle development-were higher at 3 months of age.

Inner Mongolia Albas goat coat is composed of wool and cashmere, corresponding to the primary and secondary hair follicles, respectively. The growth of cashmere in adult cashmere goats follows an annual cycle. In general, cashmere growth begins in the summer, stops in the winter, and is shed naturally in the spring; with no cashmere growing on the body surface between spring and summer. Cashmere quality is affected by genetic, breeding management and environmental factors [28,29,30,31]. With increase in age, the fleece or cashmere yield of sheep and goats showed a trend of increase and then decrease [32,33]. The cashmere diameter of Inner Mongolia and Liaoning cashmere goats tended to become thicker with age [34]. Our previous research found that cashmere production reaches a maximum at 2–5 years old and cashmere length reaches a maximum at 2–3 years old. Cashmere diameter tends to increase and then decrease, and single kids have a higher production performance than double kids. This study showed almost no change in the cashmere length and diameter of the first and second combing. Cashmere yield and unit yield decreased significantly, which was nearly consistent with the previous results.

Primary and secondary hair follicles form the hair follicle groups. These are separated by dense connective tissue surrounding the septum, each consisting of several primary follicles and a number of secondary follicles located on its side. Inner Mongolia Albas cashmere goats have mainly three-hair groups, with fewer two-hair and four-hair groups; that is, where the hair follicle group is composed of three primary hair follicles and secondary hair follicles. The development of hair follicles is a complex process that begins in the embryonic stage, and primary hair follicles occur earlier than secondary hair follicles [35]. Secondary hair follicles are derived from the epidermis surrounding the primary hair follicle-not directly from the primary hair follicle-and developed secondary hair follicles enter periodic growth and apoptosis. Fozi et al. found that the number of primary hair follicles in Raieni cashmere goats did not change after birth, and the number of secondary hair follicles stopped increasing at 3 months of age [36]. Most of the primary hair follicles of cashmere goats are completed before birth; while the number of secondary hair follicles is completed by 3–6 months after birth [37]. Moreover, Yang’s study suggests that the number of secondary hair follicles fixed during young age determines the number of secondary hair follicles in adulthood [38]. The time of primary hair follicle development in this study was later than in other studies, possibly due to the rearing environment. Secondary hair follicle results were nearly consistent with those of previous studies. In our test of skin hair follicles, the hair bulb part of the primary hair follicle was large and located in the deep layer of the skin. The diameter of the hair follicle was thick, and the hair shaft medullary. The hair bulb of the secondary hair follicle was small and located in the superficial layer of the skin. The diameter of the hair follicle was small, and the hair shaft had no medulla. Although primary and secondary hair follicles have different characteristics, they have the same basic structure. Experiments have shown that hair follicles develop from hair follicle stem cells stored in the bulge near the skin’s surface [39,40]. We found that secondary hair follicles descend to the root of primary hair follicles with increasing age, indicating that the occurrence and development of secondary hair follicles in Albas cashmere goats are similar to that of other mammals.

In fur-bearing animals, their fur is of great economic value. Cashmere production is the main economic trait of cashmere goats. The development of secondary hair follicles directly affects the yield and quality of cashmere. The greater the SFD, the more fibers and the greater the yield of cashmere. Additionally, the S:P ratio was not affected by skin shrinkage during skin tests and was generally used as an index to measure the number of secondary hair follicles in cashmere goat skin. This study showed that the PFD and SFD gradually decreased and became stable after birth, and that the S:P reached its maximum between 3 to 5 months after birth [41]. In our study, secondary hair follicles’ characteristics were unchanged at 5–6 months of age and close to the middle anagen period of the following year (1.5 years old), indicating that secondary hair follicle traits at this time are almost the same in adulthood. Therefore, hair follicle traits at the age of 6 months can be an essential indicator for breed selection.

The hair follicle is the basis for the formation and development of animal hair; and it is an organ that grows periodically. The growth cycle is regulated by a variety of signaling pathways, and *FGFs* play essential role in hair follicle morphogenesis [42,43]. Fon, T. K. et al. reported that *FGF2* and *FGF21* are expressed periodically during hair follicle growth in postnatal mice [44]. *FGF21* is mainly expressed in the inner and outer root sheaths of hair follicles, and its expression is correlated with the number of hair follicles [45,46]. *FGF*2 can promote the mitosis of dermal fibroblasts and maintain stem cells’ proliferation and differentiation ability [47]. This study found that the expression of *FGF2* and *FGF21* genes and proteins in goat kids was the highest at 3 months of age, and that the hair follicle growth rate was fastest at this time. We speculate that *FGF2* and *FGF21* may promote the development of secondary hair follicle at an early age. *BMP* determines bone formation at different stages of growth and development in animals and is one of the signaling molecules involved in hair follicle development [48,49]. As critical gene regulators, *BMPs* control the differentiation of the postnatal hair follicle and show different expression levels at various stages of hair follicle development [50]. *BMP4* is expressed and plays a negative regulatory role in the development of hair follicles and feathers and is an essential factor in the maintenance of hair follicle telogen [51,52]. *BMP*7 is expressed in the whole hair follicle cycle, with the highest expression in the growth phase, and the expression of *BMP*7 stops in the stationary phase [53]. The results of this study of *BMP4* and *BMP*7 are nearly consistent. In conclusion, the *FGF* and *BMP* genes may play an essential role in developing secondary hair follicles in goat kids.

## 5. Conclusions

We demonstrated for the first time the timing of the complete development of primary and secondary hair follicles in goat kids by serial histomorphological observations. The cashmere yield and unit yield of the first combing were significantly higher than those of the second combing. *FGF*2, *21* and *BMP*7 may promote the growth of secondary hair follicles. In addition, the period prior to six months of age is the window for the regulation of the development of secondary hair follicles and cashmere. The traits of the secondary hair follicle and cashmere at the age of 6 months can therefore be a basis for predicting cashmere quality in adulthood.

## Figures and Tables

**Figure 1 animals-13-00617-f001:**
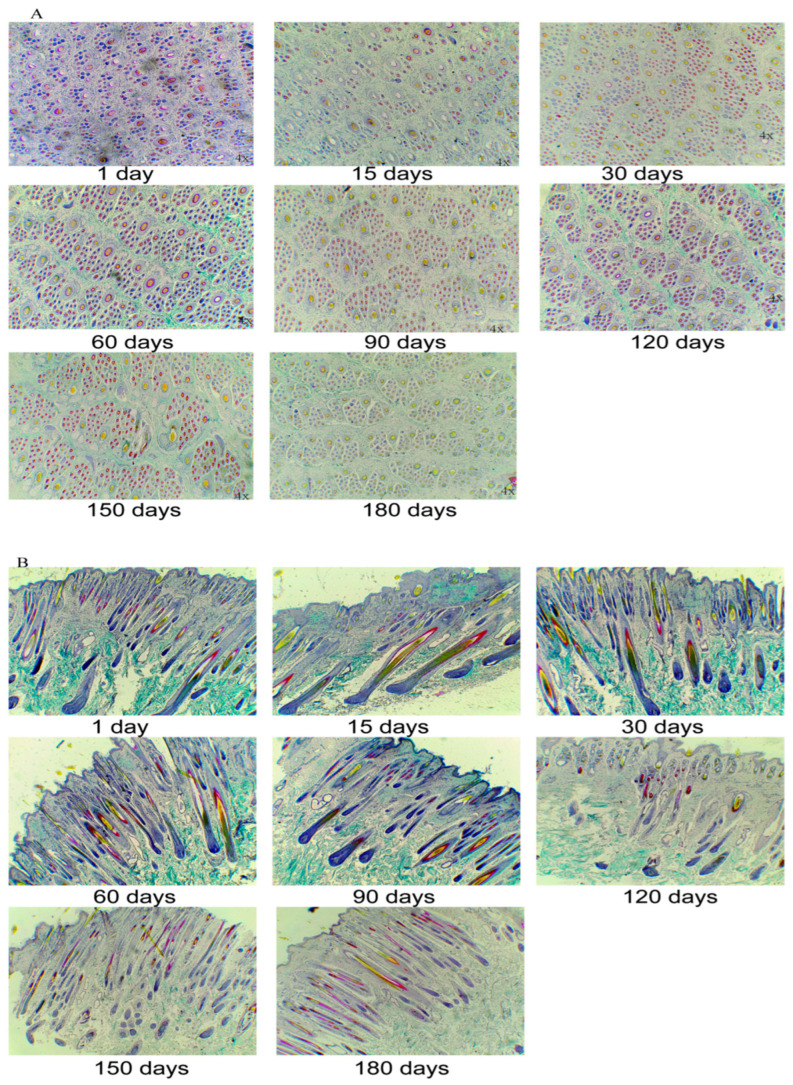
Morphological changes of hair follicles at different time points, (**A**) cross section, 40× (**B**) longitudinal section, 40×.

**Figure 2 animals-13-00617-f002:**
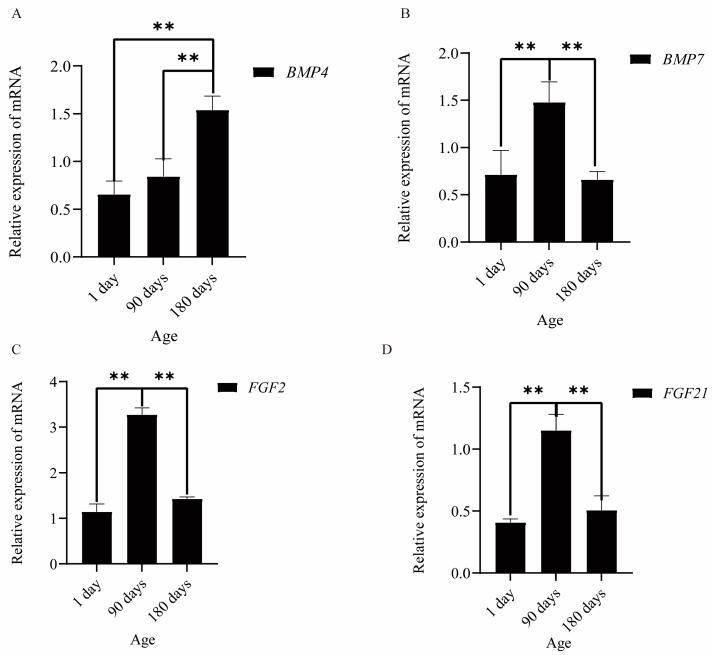
(**A**–**D**) represent the expression of *BMP4*, *BMP7*, *FGF2*, and *FGF21* at different time points (*n* = 5); Data are presented as least squares means ± SEM; ** signifies a highly significant difference at *p* ≤ 0.01.

**Figure 3 animals-13-00617-f003:**
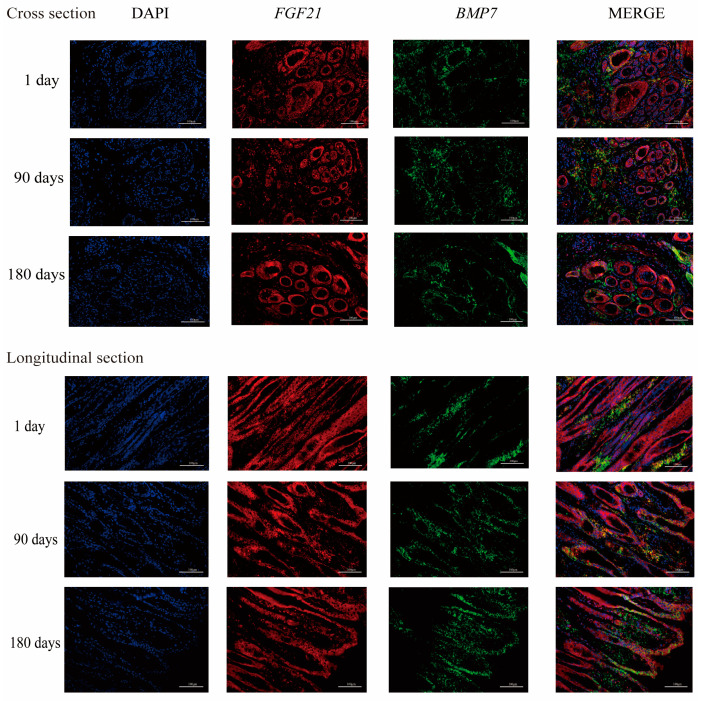
Immunofluorescence for *FGF21* and *BMP*7 in skin and hair follicle sections. Fluorescence signal DAPI (blue), *FGF21* (red), *BMP*7 (green), Scale bar = 100 μm.

**Table 1 animals-13-00617-t001:** The expression of RNA in skin samples of different ages.

Age	*FGF2*	*FGF21*	*BMP4*	*BMP7*
90 days	32.93 ± 7.11	23.48 ± 2.67	21.31 ± 3.94	16.19 ± 2.18
1 years old	1.50 ± 0.25	3.51 ± 0.23	6.16 ± 1.05	5.72 ± 3.07
6 years old	1.69 ± 0.36	5.06 ± 2.21	5.21 ± 0.84	6.06 ± 2.68

The expression quantity is the pair value base 10.

**Table 2 animals-13-00617-t002:** Primers used for expression analysis in the study.

Target	Primers Sequences (5′-3′)	Product Size (bp)
*FGF*2	F: GCAAACCGTTACCTTGCTATGA R: TACTGCCCAGTTCGTTTCAGTG	164
*FGF*21	F: GTTCAAGCACTTGGGACTGTGG R: CTGGGCATCATCCGTGTAGAG	145
*BMP*4	F: GGATTACATGCGGGATCTTTAC R: GAGTTTTCGCTGGTCCCTGG	171
*BMP*7	F: GACTTCAGCCTGGACAACGA R: TCGGTGAGGAAGTGGCTATCTT	299
GAPDH	F: ATGTTTGTGATGGGCGTGAA R: GGCGTGGACAGTGGTCATAAGT	153

GAPDH is used as control; F = forward; R = reverse.

**Table 3 animals-13-00617-t003:** Live weight at different time points.

Age	Live Weight
1 day	3.0 ^a^
15 days	5.03 ^b^
1 month	7.97 ^c^
2 months	10.09 ^d^
3 months	11.77 ^d^
4 months	13.17 ^e^
5 months	16.12 ^f^
6 months	18.52 ^f^
First yearling combing	28.61 ^g^
Second yearling combing	28.41 ^g^

First yearling combing occurs at 12 months of age, and second yearling combing at 24 months of age. ^a-g^ Different superscript letters in the same variable indicate statistical differences (*p* < 0.05).

**Table 4 animals-13-00617-t004:** Cashmere performance at different time points.

Item	3 Months	6 Months	First Yearling Combing	Second Yearling Combing
Diameter/μm	13.40 ^a,b^	12.80 ^a^	13.65 ^b^	13.39 ^a,b^
Length/cm	2.20 ^a^	4.57 ^b^	9.09 ^c^	9.11 ^c^
Yield/g	-	-	869.48 ^b^	695.73 ^a^
Production unit weight g/kg	-	-	30.58 ^b^	24.54 ^a^

^a–c^ Different superscript letters in the same variable indicate statistical differences (*p* < 0.05).

**Table 5 animals-13-00617-t005:** The hair follicle trial at different time points.

Age	PFD (n/mm²)	SFD (n/mm²)	MSFD (n/mm²)	Ratio of MSF	S:P
Day 1	9.97 ^e^	60.75 ^e^	30.33 ^e^	0.500 ^a^	6.10 ^a^
Day 15	9.37 ^e^	57.81 ^d,e^	31.01 ^e^	0.534 ^a^	6.10 ^a^
Month 1	8.52 ^d^	57.56 ^d,e^	35.71 ^e,d^	0.619 ^b^	6.87 ^b^
Month 2	6.80 ^c^	55.80 ^d^	39.06 ^d^	0.700 ^c^	8.23 ^c^
Month 3	5.57 ^c^	54.52 ^c,d^	44.86 ^c^	0.823 ^d^	9.80 ^d^
Month 4	4.52 ^b^	53.78 ^c^	49.03 ^b^	0.911 ^e^	11.95 ^e^
Month 5	4.41 ^b^	51.57 ^b^	50.41 ^b^	0.971 ^f^	11.74 ^e^
Month 6	3.86 ^a^	48.68 ^a^	48.68 ^a^	1.000 ^f^	12.64 ^e^
1.5 years	4.01 ^a^	46.83 ^a^	46.83 ^a^	1.000 ^f^	11.82 ^e^

PFD = primary hair follicle density; SFD = secondary hair follicle density; MSFD = mature secondary hair follicle density; Ratio of MSF = the ratio of the number of mature secondary follicles to the total number of secondary follicles; S:P = primary hair follicle/secondary hair follicle. ^a–f^ Different superscript letters in the same variable indicate statistical differences (*p* < 0.05).

## Data Availability

The data used to support the findings of this study have not been made available as we will be conducting further research.

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
