# Peer review of "Hair Follicle Development and Cashmere Traits in Albas Goat Kids"

_animals, 2023, doi:10.3390/ani13040617_

Round 1

Reviewer 1 Report

The manuscript describes a longitudinal study of skin follicle development in cashmere goats sampled on 9 occasions from day 1 to 2 years of age.  Results included measurements of gene expression and of their peptide products for BMP and FGF conducted on 3 occasions; days 1, 90 and 180. This limited number of observations of gene expression makes it difficult to reconcile the title of the paper with the results and seriously limits the strength of any conclusions about these genes having a ‘vital role’ in the development of secondary hair follicles. For these reasons the manuscript requires some revision to make it suitable for publication.  Also, the standard of English is poor in places and needs some editorial input before it reaches publication quality.  The study has been well conducted and the findings are adequately presented and merit publication, but it has to be conceded more strongly than it has been that the gene expression measurements are too few to enable any more to be claimed than a faint association with the follicular information.

In its current form the title of the manuscript does not make sense. Also, mention of FGF and BMP expression in the title does not describe the main purpose and findings of the study. It should be written so it is clear this is a longitudinal study of hair follicle development with minimal observations of gene expression.  The statement in section 3.3 that states 2 months of age is ‘the critical period of secondary hair follicle development’ is provided without a source reference, unless the authors are claiming this on the basis of data presented here for which proof is lacking.  All that can be said about the gene expression and protein data is that expression was higher at day 90 than on day 1 and, in 3 out of 4 cases, had dropped at day 180.  It must be stressed that this provides little in the way of evidence for a critical role of the genes in the development of secondary fibres.  Other points to note:  it should be ‘live weight’ not ‘body weight’ throughout the text; it should be ‘ratio’ not ‘rate’ in Table 4 and ‘MSF’ needs to be defined (it is the ratio of the number of mature secondary follicles to the total number of secondary follicles); in 5. Conclusions it should be ‘histomorphological’ not ‘histopathological’; the tables could benefit from clearer layout; some of the references need attention – e.g. 5 and 18.

Author Response

Response: Thank you very much for reviewing our manuscript. Your comments are valuable and very helpful for revising and improving our paper. We have studied comments carefully and revised the manuscript according to your detailed suggestions. Revised portion are marked in the paper. The response are as follows:

1 we changed the title to “The Study on the hair follicle development and the cashmere traits in Albas goat kids”, so that the results of article are more consistent with the title;

2 We deleted 'the critical period of secondary hair follicle development'. The growth of secondary hair follicles in adult cashmere goats has obvious periodicity, including anagen (active), catagen (quiescent) and telogen (inactive) phases. Hair follicle development begins in the embryonic period, and secondary hair follicles occur later than primary hair follicles. After kids are born, the secondary hair follicles are still growing and developing. It is found that the change of MSFD is maximum from 2 to 3 months in this study, and the expression of FGF2/21 and BMP7 was the highest at this time. Therefore, we speculate that FGF2/21 and BMP7 play key role in the growth of secondary hair follicles and add this content in the discussion;

3 The Introduction is redesigned with a brief content about the FGF and the BMP;

4 We added a description of live weight to Results3.1;

5 The parts of this article that need to be modified have been revised according to the comments of your advice;

6 we modified the sentence and syntax of the full text and marked it in the article;

7 The format of all references has been modified as requested;

The above is my response. I hope my answers will satisfy you. Thank you again for your review.

Reviewer 2 Report

Why do you directly select FGF2, FGF21, BMP4, and BMP7 for PCR instead of other members of the FGF family, and how do you make sure that these four genes are present in your sample? There are some problems in the structure of the article, and it is suggested to redesign.

Author Response

 Why do you directly select FGF2, FGF21, BMP4, and BMP7 for PCR instead of other members of the FGF family, and how do you make sure that these four genes are present in your sample? There are some problems in the structure of the article, and it is suggested to redesign.

Response: Thank you very much for reviewing our manuscript. Your comments are valuable and very helpful for revising and improving our paper. We have studied comments carefully and revised the manuscript according to your detailed suggestions. Revised portion are marked in the paper. The response are as follows:

First, The ‘Introduction’ is redesigned with a brief content about the FGF and the BMP. Secondly, FGF2, FGF21, BMP4 and BMP7 were selected as the target genes based on our previous analysis of the RNA-seq data of goat kids and adult goats. And, most studies about FGF2/21 and BMP4/7 focused on the adult cashmere goats, few studies conducted on kids. Finally, we changed the title to “The Study on the hair follicle development and the cashmere traits in Albas goat kids”, so that the results of article are more consistent with the title.

In addition, we added a description of live weight to the Results3.1 and modified the sentence and syntax of the full text and marked it in the article. I hope my answers will satisfy you. Thank you again for your review.

Round 2

Reviewer 2 Report

There are some problems in the structure of the article. The first problem was not modified. Authors need to carefully design experiments to achieve the rationality of the results of the article

Author Response

Response:Thank you very much for your attention and the reviewer’s comments on our manuscript. Those comments are valuable and very helpful for revising and improving our paper. We have studied comments carefully and revised the manuscript according to your kind advice and reviewer’s detailed suggestions. Revised portion (Red) are marked in the paper.

This thesis is a long-term tracing experiment in which live weight, cashmere and hair follicle traits were recorded and analysed from goat kids to adult. In addition, the expression of four key genes and proteins in the skin follicles of kids was examined. Finally, we conclude that the completion time of primary and secondary hair follicle development, as well as the expression pattern of FGF2/21 and BMP4/7. What ‘s more, the secondary hair follicle traits and the cashmere traits at 6 months can be used as indicators for species selection of cashmere goats. We made the following changes to this article.

  1. RNA-seq data for FGF and BMP at different ages of 90 days old, 1 year and 6 years added to the article.
  2. with the addition of data on weight at different stages.
  3. A comparative analysis of the different stages of hair follicle traits was carried out. This better highlights the purpose of our experiment and the title of the thesis.
  4. The grammar in the article has been touched up.

We hope our response satisfy you.